# Molecular architecture of the assembly of *Bacillus* spore coat protein GerQ revealed by cryo-EM

Yijia Cheng [1,3], Mark A. B. Kreutzberger[2,3], Jianting Han [1], Edward H. Egelman [2] ✉ & Qin Cao [1] ✉

Protein filaments are ubiquitous in nature and have diverse biological functions. Cryo-electron microscopy (cryo-EM) enables the determination of atomic structures, even from native samples, and is capable of identifying previously unknown filament species through high-resolution cryo-EM maps. In this study, we determine the structure of an unreported filament species from a cryo-EM dataset collected from *Bacillus amyloiquefaciens* biofilms. These filaments are composed of GerQ, a spore coat protein known to be involved in *Bacillus* spore germination. GerQ assembles into a structurally stable architecture consisting of rings containing nine subunits, which stacks to form filaments. Molecular dockings and model predictions suggest that this nine-subunit structure is suitable for binding CwlJ, a protein recruited by GerQ and essential for $Ca^{2+}$-DPA induced spore germination. While the assembly state of GerQ within the spores and the direct interaction between GerQ and CwlJ have yet to be validated through further experiments, our findings provide valuable insights into the self-assembly of GerQ and enhance our understanding of its role in spore germination.

Bacterial endospores are a distinct cellular form characterized by metabolic dormancy and high resistance to harsh treatments[1]. Through the process of sporulation, bacteria can survive extreme environmental conditions by encapsulating their genome within the spores[2]. Once environmental conditions become favorable again, the spores can rapidly germinate and transition back to the vegetative state. The spores are enveloped by a proteinaceous, self-assembled, multilayered structure known as the spore coat. Extensive studies on the spore coat of *Bacillus subtilis* have revealed its composition, comprising over 70 proteins[2]. These spore coat proteins play crucial roles in spore coat assembly and anchoring, spore cortex formation, protection against harsh conditions, and germination[3]. Despite the significance of these spore coat proteins, there is limited structural information available regarding their assembly mechanism. Since bacterial spores, such as those produced by *Bacillus anthracis* and

*Clostridium difficile*, can play an important role in human diseases[4], understanding the structure and assembly of the spore coat has great clinical significance.

Protein filaments are ubiquitous across all life and serve important functions in various biological processes, such as cytoskeleton in cell integrity and migration[5], zona pellucida (ZP) module filaments in hearing, fertilization, and antibacterial defense[6], and amyloid fibrils in human diseases[7]. Understanding the molecular architecture of these filaments is crucial for unraveling their functional mechanism. Cryo-electron microscopy (cryo-EM) has proven to be a powerful technique for determining the structure of protein filaments from native biological samples. In addition, it is capable of identifying previously unreported protein filaments from high-resolution cryo-EM maps[8–11]. In a previous study, we identified filaments formed by the extracellular peptidase Vpr in *B. amyloiquefaciens* biofilm using cryo-EM[11]. Within

[1]Bio-X Institutes, Key Laboratory for the Genetics of Developmental and Neuropsychiatric Disorders, Ministry of Education, Shanghai Jiao Tong University, Shanghai 200030, China. [2]Department of Biochemistry and Molecular Genetics, University of Virginia School of Medicine, Charlottesville, VA 22903, USA. [3]These authors contributed equally: Yijia Cheng, Mark A. B. Kreutzberger. ✉e-mail: egelman@virginia.edu; caoqin@sjtu.edu.cn

the cryo-EM dataset, we also observed another filament species characterized by bundled 8-nm-wide filaments[11]. However, the structure of this filament species was not determined in the previous study due to its bundled nature.

In this study, we re-process the previous cryo-EM data and successfully determine the structure of these 8 nm filaments at a resolution of 3.3 Å. Atomic model building reveals that these filaments are composed by GerQ, a spore coat protein found in the inner layer of the spore coat. GerQ plays a crucial role in the germination of *Bacillus* spores[12,13]. Our findings demonstrate that GerQ self-assembles into filaments under physiological conditions. This discovery provides valuable insights for investigating the functional roles of GerQ in *Bacillus* spore assembly, resistance, and germination.

## Results

### Cryo-EM structure determination of the 8 nm filament

In our previous cryo-EM dataset, we observed 8 nm filaments primarily in a bundled form (Fig. 1a, grey arrow), alongside flagella and Vpr filaments (Fig. 1a, blue and green arrows). To determine the structure of these 8 nm filaments, we manually picked particles from partially unbundled filaments (as indicated in Fig. 1a, cyan arrow) and used the resulting two-dimensional (2D) classes as templates for automatic particle picking (Supplementary Fig. 1). Among the 2D classes generated from the automatically picked particles, we identified one class that appeared unbundled (Fig. 1b, top panel), while the major of the classes still bundled (Fig. 1b, bottom panel). Both bundled and unbundled 2D classes displayed similar filament morphology,

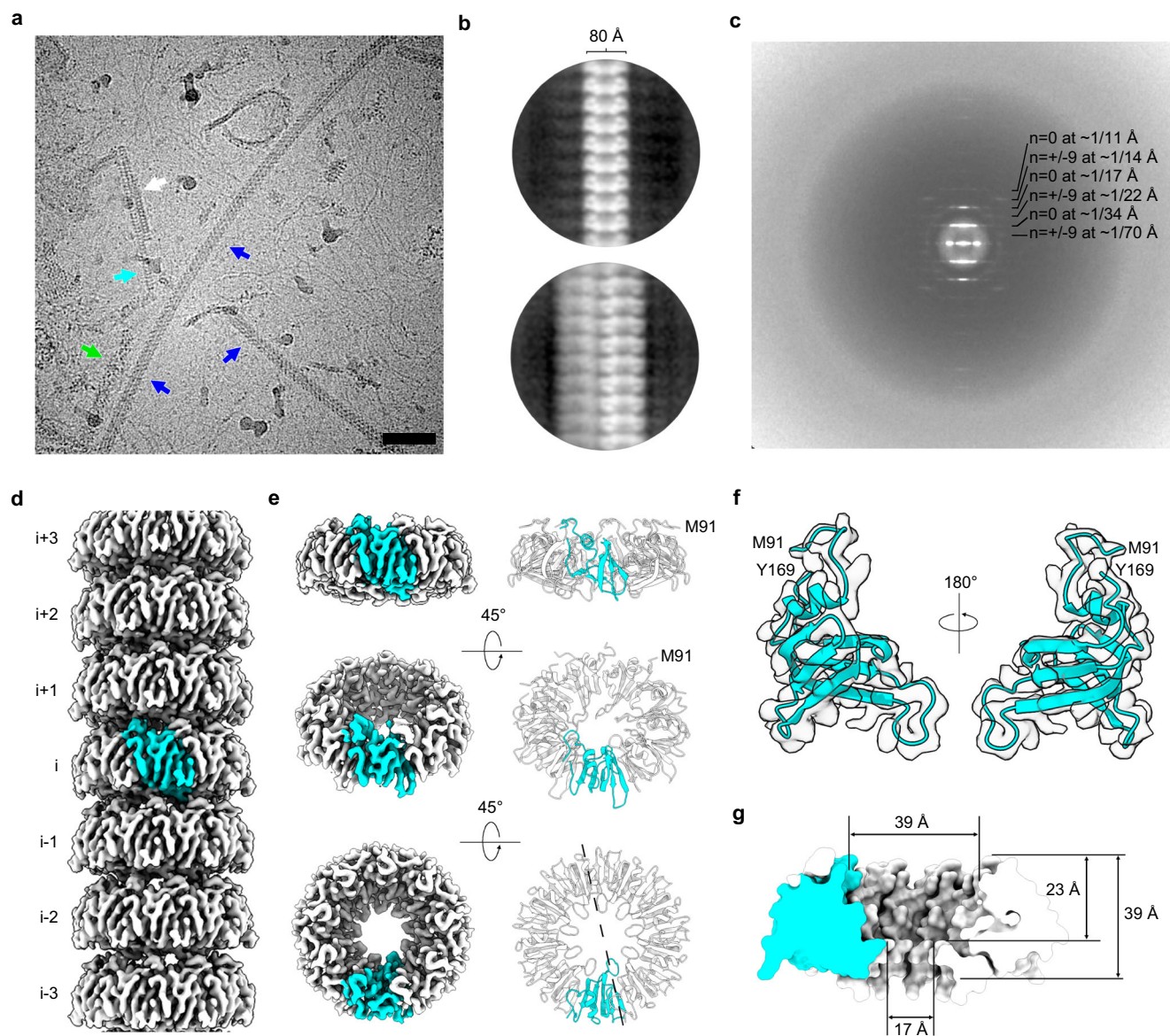

**Fig. 1 | Cryo-EM structure of the GerQ filament. a** Representative cryo-EM micrograph of the previously collected dataset (contain a total of 2894 micrographs). Filaments previously identified as flagella (blue), Vpr (green), and bundled 8 nm filaments (grey) are indicated by arrows. The partially unbundled 8 nm filament used in cryo-EM data processing in this study is indicated by cyan arrow. Scale bar = 50 nm. **b** Two-dimensional (2D) classes of unbundled (top) and bundled (bottom) 8 nm filament. **c** Power spectrum of the GerQ filament. There are two apparent kinds of layer lines present corresponding to Bessel functions of $n = 0$ or overlapping Bessel functions of $n = +9$ and $n = −9$. **d** The cryo-EM map of the GerQ filament with a single asymmetric unit colored in cyan. **e** The cryo-EM map (left panels) and atomic model (right panels) of one layer of the GerQ filament. The dashed line in the bottom right panel indicates the direction of the split view displayed in (**g**). **f** The cryo-EM map and atomic model of one subunit of the GerQ filament. **g** The split view of one layer of the GerQ filament, shown as Van der Waals surface.

**Table 1 | Cryo-EM data collection, refinement and validation statistics of GerQ filaments**

| | GerQ (EMD-60394, PDB 8ZRA) |
|---|---|
| **Data collection and processing** | |
| Magnification | ×130,000 |
| Voltage (kV) | 300 |
| Electron exposure (e⁻/Å2) | 40 |
| Defocus range (µm) | 0.6-4.6 |
| Pixel size (Å) | 1.05 |
| Symmetry imposed | $C_9$ |
| Helical rise (Å) | 34.2 |
| Helical twist (°) | −18.4 |
| Initial particle images (no.) | 729,732 |
| Final particle images (no.) | 17,798 |
| Map resolution (Å) | 3.3 |
| FSC threshold | 0.143 |
| Map resolution range (Å) | 200-3.3 |
| **Refinement** | |
| Initial model used (PDB code) | De novo |
| Model resolution (Å) | 3.5 |
| FSC threshold | 0.5 |
| Model resolution range (Å) | 200-3.5 |
| Map sharpening *B* factor (Å2) | −96 |
| Model composition | |
| Nonhydrogen atoms | 11,772 |
| Protein residues | 1422 |
| Ligands | 0 |
| *B* factors (Å2) | |
| Protein | 75.2 |
| Ligand | – |
| R.m.s. deviations | |
| Bond lengths (Å) | 0.006 |
| Bond angles (°) | 1.137 |
| **Validation** | |
| MolProbity score | 2.29 |
| Clashscore | 16.83 |
| Poor rotamers (%) | 0 |
| Ramachandran plot | |
| Favored (%) | 89.6 |
| Allowed (%) | 10.4 |
| Disallowed (%) | 0 |

indicating that they belong to the same filament species. We selected particles from the unbundled class for further 3D reconstruction, employing potential symmetries determined from the power spectrum analysis of a 2D class (Fig. 1c, see Methods). Employing $C_9$ symmetry and a helical symmetry, we obtained a cryo-EM map of these 8 nm filaments at a resolution of 3.3 Å (Fig. 1d, Supplementary Fig. 2). Detailed data collection and processing statistics are presented in Table 1.

### Atomic model building and identification of the filament protein

The identity of the filament protein was determined using a previously described method[11]. An initial model was automatically built using ModelAngelo[14] without providing a starting protein sequence. The predicted amino acid sequence of the initial model was then used in a BLAST search to find actual proteins with similar sequences[15,16]. The search results revealed that the spore coat protein GerQ (also known as YwdL) from various *Bacillus* species displayed the highest sequence

identity among the hits. Genotyping was performed to identify the GerQ sequence in the bacteria strain utilized to prepare the cryo-EM sample[11], and the genotyped sequence was used to build the final model of the filament. Both the final model and the initial ModelAngelo model fit the map well, with most side chain densities being well explained by the models (Supplementary Fig. 3a, b). Additionally, the final model is consistent with the AlphaFold 3 predicted model of GerQ[17] (Supplementary Fig. 3a). These observations support the identity of the filament protein as GerQ.

### Structure of GerQ filaments

The GerQ filament exhibits an overall structure characterized by stacked layers, with each layer composed of nine GerQ subunits that form a bowl-like shape (Figs. 1d, e and 2a). These layers are related to each other by helical symmetry with a rise of 34.2 Å and a twist of −18.4°. Within each layer, the nine GerQ subunits are related to each other with $C_9$ point group symmetry. The filament core in a single GerQ subunit contains residues 91-169. It comprises two α-helixes, five β-strands, and connecting loops that link these secondary structures (Figs. 1f and 2b, Supplementary Fig. 3 & 4). The bowl-like structure formed by the nine subunits features a wide opening (~39 Å) on one side and a narrow opening (-17 Å) on the other side. Additionally, the wide opening creates a chamber approximately 23 Å deep, which is covered by an additional layer stacked on top (Fig. 1g).

The bowl-like structure of the GerQ filament is assembled through nine symmetrically related interaction interfaces, exhibiting a $C_9$ symmetry (Fig. 2a). Each interface is formed by two adjacent GerQ subunits, where β-strands from both GerQ subunits create a continuous antiparallel β-sheet (Fig. 2b). The subunit-to-subunit interface is further stabilized by hydrophobic interactions and π-π stackings. Key residues involved in these interactions include Phe117, Trp123, Phe128, Ile141, Leu152, and Leu154 from one subunit, as well as Ile99, Ile102, Tyr114, Leu158, Tyr160, Phe163, and Ile167 from the other subunit (Fig. 2c). Notably, in the absence of the subunit-to-subunit interface, the isolated GerQ monomer would expose numerous hydrophobic surfaces, rendering it unstable in solution (Fig. 2d).

In contrast to the tightly packed bowl-like structure, the interactions between two adjacent layers in the GerQ filament are relatively weak. The layer-to-layer interface primarily involves hydrophilic interactions, including a long-distance salt bridge between Glu96 and Glu95 of the bottom layer and Arg129 of the top layer, another long-distance salt bridge between Arg137 of the bottom layer and Glu122 of the top layer, as well as a hydrogen bond network involving Tyr98 of the bottom layer and Glu122 and Lys121 of the top layer (Fig. 2e). Due to the $C_9$ symmetry, there are a total of nine sets of the aforementioned interactions that connect two adjacent layers. These interactions are positioned around the perimeter of the wide opening of the bottom bowl-like structure (Fig. 2e).

### GerQ filaments naturally existed outside the *B. amyloiquefaciens* spores

To investigate whether the GerQ filaments observed were naturally produced by *B. amyloiquefaciens* or artificially induced during filament extraction, we conducted experiments using *B. amyloiquefaciens* cultures grown under the same conditions as the sample used for cryo-EM data collection (on YESCA plates, see Methods). We grew the bacteria for different durations, gently scraped them from the plates, mixed them with buffers, and directly applied the resulting solution onto negative stain EM grids for observation. We observed the presence of GerQ filaments in cultures that were grown for 3 days or longer, while these filaments were absent in 1-day cultures (Fig. 3, left panels). The appearance of GerQ filaments seemed to coincide with the emergence of the spores, as indicated by malachite green staining (Fig. 3, left panels). These findings suggest that the GerQ filaments naturally exist

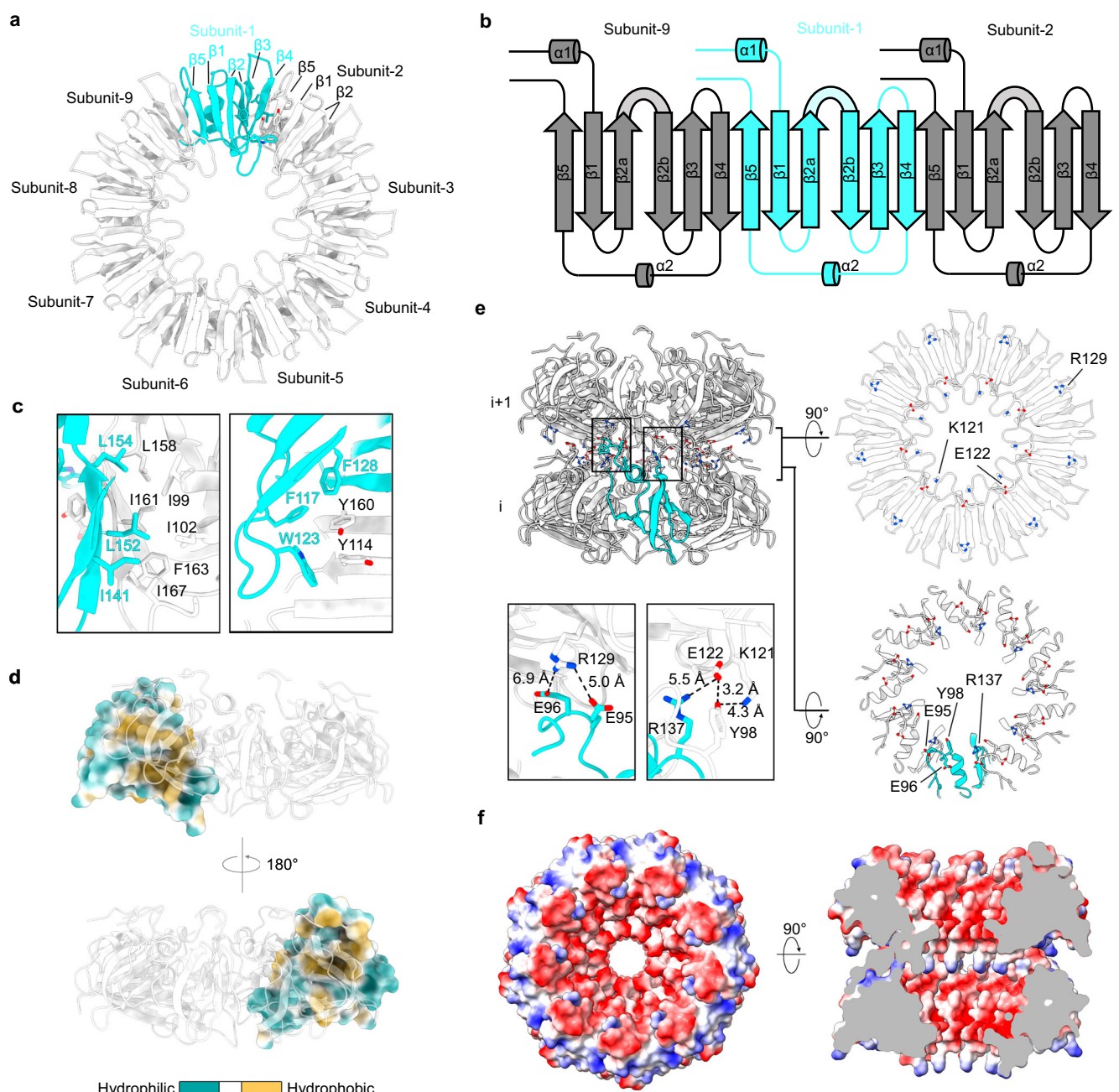

**Fig. 2 | Structure analysis of the GerQ filaments. a** Top view of a single layer of the GerQ filament, composed of 9 subunits (subunit-1 to −9). Subunit-1 is colored in cyan and the other subunits are colored in white. **b** Topological diagram of three adjacent subunits in a single layer of the GerQ filament. **c** Detailed structures of the interface between subunit-1 (cyan) and subunit-2 (white). **d** Hydrophobicity analysis of the subunit-to-subunit interfaces. One layer of the GerQ filament is shown as the side view, with one subunit represented as surface and colored based on the molecular lipophilicity potential (ranging from dark cyan for the most hydrophilic to goldenrod for the most hydrophobic). The other subunits are represented as cartoons, colored in white and shown as transparent. **e** Analysis of the layer-to-layer interface. The top left panel shows a side view of two layers of the GerQ filament. One subunit is colored in cyan, while the remaining subunits are white. Residues involved in the layer-to-layer interaction are shown as sticks. Two hydrophilic interaction networks are indicated with frames, with details shown in the bottom left panels. The bottom view of the upper layer and the top view of the lower layer are shown in the right panels. **f** Electrostatic diagrams of two layers of the GerQ filament.

in *B. amyloiquefaciens* cultures and that the formation of these filaments may be associated with sporulation process.

Additionally, these findings also indicate that the observed GerQ filaments, as visualized under negative stain EM, are located outside the spores and vegetative cells, because we did not employ any treatment that could potentially disrupt the spores or bacterial cells during the sample preparation. To explore the presence of these extracellular GerQ filaments under different culture conditions or in cultures of other *Bacillus* bacteria, we conducted additional

experiments. Specifically, we grew *B. amyloiquefaciens* in 2 × SG liquid medium and *B. Subtilis* strain 168 on YESCA plates. Despite the observation of sporulation in both cultures, as confirmed by malachite green staining (Supplementary Fig. 5a), the GerQ filaments were not detected under negative stain EM. We also cultured purified *B. amyloiquefaciens* spores (as described in the following paragraph) under a nutrient-rich condition to induce germination, and no GerQ filaments were observed outside these germinated spores (Supplementary Fig. 5b). These results suggest that these extracellular GerQ filaments

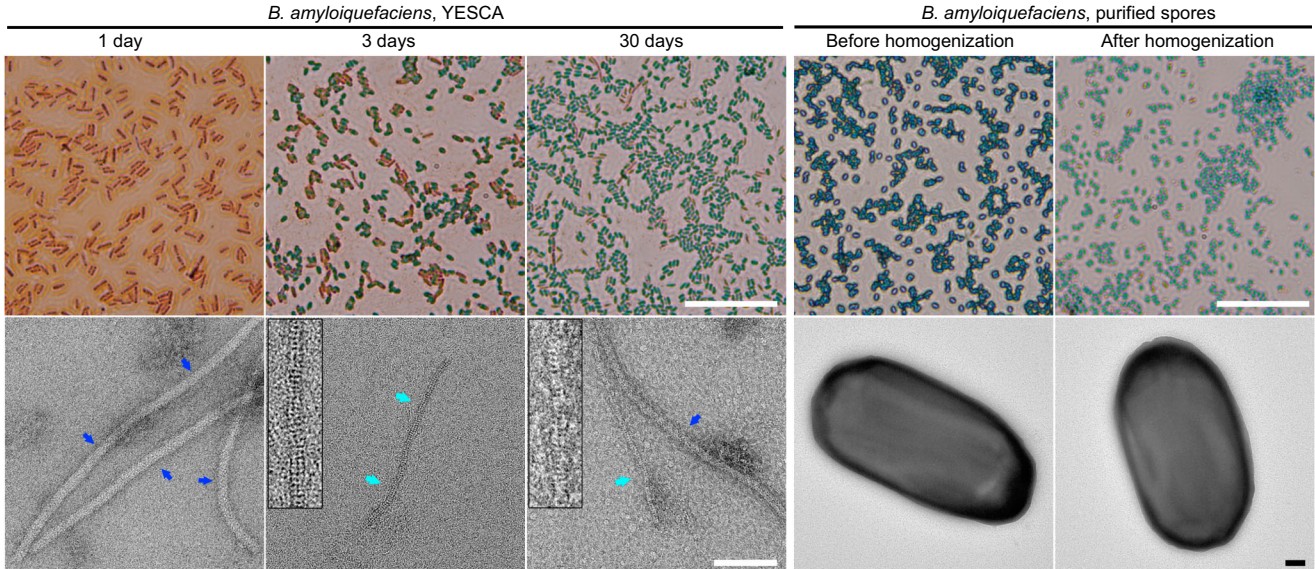

**Fig. 3 | Sporulation and GerQ filament formation in *B. amyloiquefaciens*.** Bacteria were grown on YESCA plates for different durations (left panels) or grown on 2 × SG plates and subsequently used for endospore purification (right panels). Malachite green staining was performed on each sample (top panels, scale bar = 50 μm), with endospores stained green and bacteria stained red. The negative stain TEM images of each sample are displayed on the bottom panels (scale bar = 100 nm). The 15 nm filaments previously identified as flagella or Vpr filaments are indicated by blue arrows, while the 8 nm filaments possessing the same morphology as the GerQ filament are indicated by cyan arrows (enlarged filaments are displayed as inserts). All experiments have been repeated independently more than three times with similar results.

may be specific to certain bacteria strains and are influenced by the particular growing conditions.

### Attempts to identify GerQ filaments within the spore coat

Previous studies have indicated that GerQ localizes in the inner coat of *B. subtilis* spores[12,18] (see Discussion). To investigate whether GerQ within the spores also forms filaments, we purified spores from *B. amyloiquefaciens* cultured in 2 × SG liquid medium using a well-established protocol[19]. The purified spores were subsequently stained with malachite green, and all particles exhibited green staining, indicating their purity and intactness (Fig. 3, right panels). We proceeded to homogenize the purified spores in a manner similar to our previous cryo-EM sample preparation[11]. Following homogenization, we observed a small fraction of spores exhibit signs of cracking, as evidenced by a change in malachite staining from green to red. However, under EM examination, neither before nor after homogenization, did we observe any GerQ filaments. Instead, only intact spores were observed (Fig. 3, right panels). These results suggest that if GerQ filaments do exist within the spores, they cannot be released through partial cracking of the spores. Subsequently, we utilized a low-temperature grinding technique on the purified spores, a method commonly employed to disrupt robust particles such as plant seeds. Post-grinding, fractured spores were visualized under EM (Supplementary Fig. 5c, left panel), indicating that the grinding process effectively ruptured the spores, potentially releasing inner coat proteins. Despite this, GerQ filaments were not detected under this condition. Instead, we observed ring-like particles measuring approximately 8 nm in width, resembling the size and shape of the bowl-like structure of GerQ (Supplementary Fig. 5c). Due to the constraints of EM resolution, definitively confirming whether these particles are GerQ is challenging (see Discussion). Finally, we generated ultrathin sections from the purified *B. amyloiquefaciens* spores, and GerQ filaments were not identified within the sections of spore coats (Supplementary Fig. 5d), likely due to the dense environment and noisy background inherent to spore coat sections. Taken together,

our attempts suggested that, at the current stage, we are unable to confirm whether GerQ forms filaments within the spores (see Discussion).

### Molecular docking and AlphaFold 3 prediction of the GerQ-CwlJ complex

GerQ has been reported to function by recruiting CwlJ, a protein responsible for hydrolyzing the cortex peptidoglycan surrounding dormant spores and essential for $Ca^{2+}$-DPA induced spore germination[12]. However, it remains unclear whether this recruitment is achieved through a direct interaction between GerQ and CwlJ. The bowl-like structure formed by the nine GerQ subunits, with its wide-open and deep chamber, suggests that GerQ may serve as a scaffold for interacting with other macromolecules. To investigate the suitability of this scaffold for interaction with CwlJ, we performed molecular docking using the GerQ filament structure determined in this study (comprising two layers of the bowl-like structure) and a predicted model of CwlJ generated by AlphaFold 3[17]. We employed two docking software, LightDock[20] and HDock[21], both of which yielded similar predictions, indicating that CwlJ potentially binds inside the bowl-like structure of GerQ (Fig. 4a, Supplementary Fig. 6). Additionally, we used AlphaFold 3 to predict a complex model consisting of nine copies of GerQ and one copy of CwlJ, without the provided structure information of GerQ (Supplementary Fig. 7). The resulting model was consistent with the molecular docking predictions (Fig. 4a, Supplementary Fig. 6). Based on these predicted models, we observed that the size of CwlJ fits well within the opening chamber of GerQ's bowl-like structure (Fig. 4b). Furthermore, the interaction between GerQ and CwlJ was primarily stabilized by coulombic forces, as the inner surface of the GerQ's bowl-like structure carries negative charges, while the corresponding surface of the CwlJ bears positive charges (Figs. 2f and 4c).

### Discussion

In this study, we determined the cryo-EM structure of an unidentified filament species from a previously reported cryo-EM dataset. The structure of this filament species was not determined in the previous

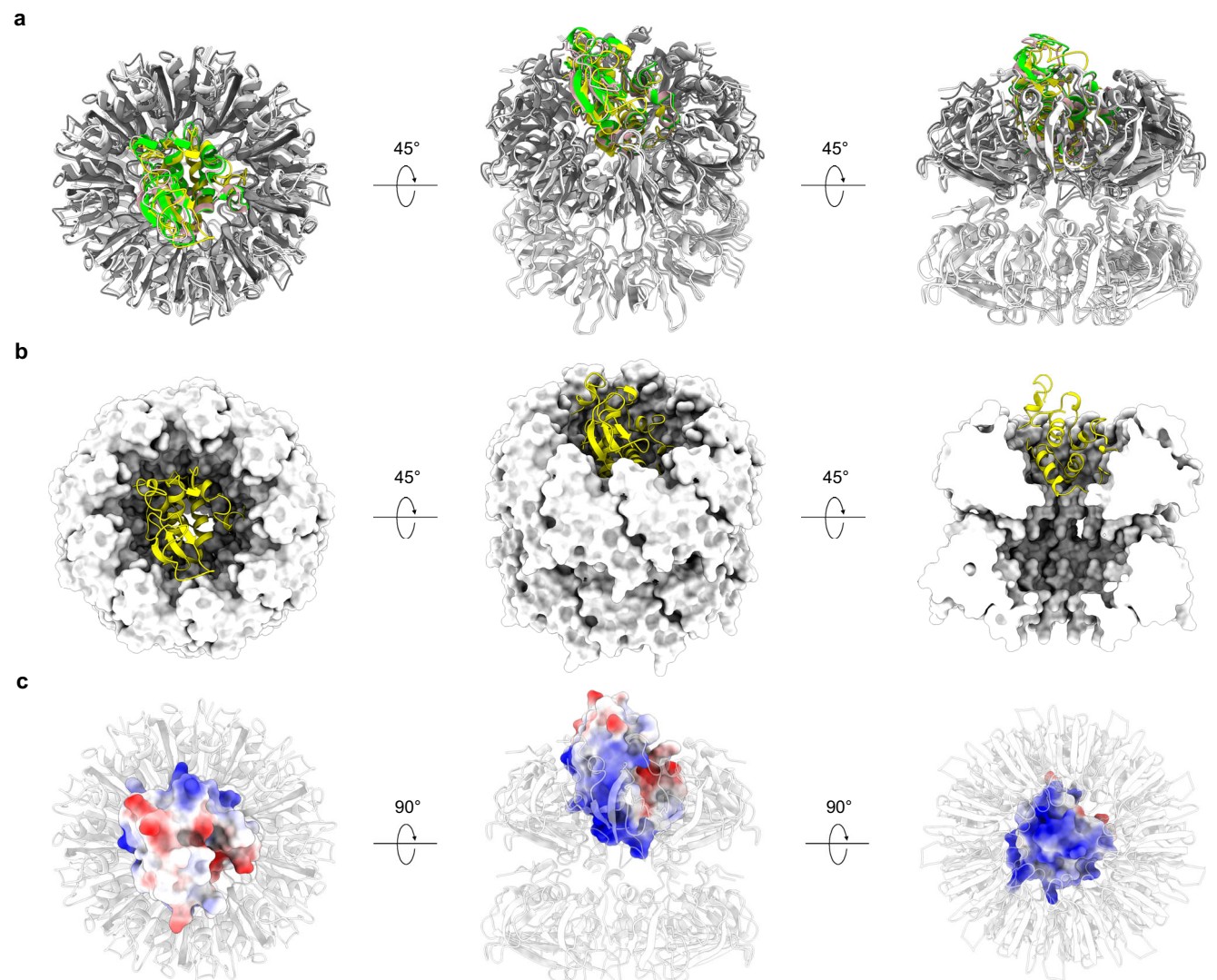

**Fig. 4 | Macromolecular docking and AlphaFold 3 prediction of the GerQ-CwlJ complex. a** Superimposition of complex models predicted by LightDock, HDock, and AlphaFold 3. GerQ is colored in white (LightDock and HDock model) or grey (AlphaFold 3 model), while CwlJ is colored yellow (LightDock), green (HDock) or pink (AlphaFold 3). Both GerQ and CwlJ are displayed as cartoons. **b, c** The complex model predicted by LighDock. GerQ is displayed as surfaces (**b**) or as cartoons (**c**) and colored in white, while CwlJ is displayed as cartoons (**b**) or as surfaces (**c**) and colored in yellow (**b**) or by coulomb potential (**c**). GerQ in (**c**) is shown in transparent.

study due to the majority of filaments being bundled and unsuitable for cryo-EM structure determination. However, through the implementation of an optimized particle picking strategy and accurate predictions of possible filament symmetries, we were able to achieve structure determination using a small fraction of partially unbundled filaments. Subsequent atomic modeling allowed us to identify these filaments as GerQ filaments. GerQ is a spore coat protein known for its importance in spore germination in *B. subtilis* and *B. cereus*[12,13]. Nevertheless, the filament assembly of GerQ has not been previously reported. Our study provides valuable molecular insights into GerQ filament formation and contributes to the functional understanding of GerQ.

Although the GerQ filaments observed in this study were located outside the spores (as discussed in the next paragraph), our structural analysis suggest that GerQ may also form the same assembly within the spores, potentially carrying out its function in this conformation. The GerQ filaments were composed of repeated stacking of bowl-like structures, with each bowl-like structure consisting of nine GerQ subunits (Fig. 1d, e). The nine-subunit assembly exhibited remarkable stability, with all subunits connected through β-sheet completion and

hydrophobic interactions (Fig. 2a–c). In comparison, the monomeric form of GerQ is expected to be much less stable due to the exposure of hydrophobic surfaces (Fig. 2d). These observations suggest that the nine-subunit assembly is likely the natural conformation of GerQ in physiological environments, including within the spores. Meanwhile, the stacking of the nine-subunit assemblies in GerQ filaments is not as stable as the assembly of the individual nine-subunit structures, primarily relying on hydrogen bonds and long-distance salt bridges. Therefore, it is inferred that the stacking of the nine-subunit assembly into filaments may not be essential for the function of GerQ within the spores. Molecular docking and AlphaFold 3 prediction suggest direct interactions between the nine-subunit GerQ assembly and CwlJ, a protein previously reported to be recruit by GerQ during spore germination. All three prediction methods generate similar complex models, indicating a specific and stable interaction between GerQ and CwlJ. These observations suggest that GerQ may recruit CwlJ by providing a structural anchor for CwlJ binding. Considering the opening of the predicted CwlJ binding chamber is blocked by the stacking of an additional layer of bowl-like structures in filament formation, it is plausible that the filament formation serves as a mechanism to protect

the CwlJ binding site on GerQ when CwlJ is absent or when the stoichiometry of CwlJ to GerQ is low. When GerQ needs to bind CwlJ, CwlJ binding may compete with the layer stacking in GerQ filaments, releasing more bowl-like structures with free binding sites. However, it is important to note that the interaction between GerQ and CwlJ is currently hypothetical and lacks experimental evidence. Efforts to identify CwlJ densities at the tip of the GerQ filament proved challenging due to the limited number of ends of the GerQ filaments within our cryo-EM dataset (see Methods). Therefore, it is still unknown whether GerQ recruits CwlJ within the spores thought direct interaction. Further experimental studies are required to validate the interaction of GerQ and CwlJ, as well as to confirm the assembly states of GerQ within the spores. Intriguingly, ring-like particles resembling the bowl-like structure of GerQ were released from low-temperature ground spores (Supplementary Fig. 5c). Although the composition of these particles as GerQ cannot be definitively confirmed, their observation provides partial support for our hypothesis that the bowl-like structure of GerQ exists within the spores, potentially representing GerQ's functional state.

The cryo-EM dataset re-processed in this study was collected from samples extracted from *B. amyloiquefaciens* biofilms, which most likely represents the extracellular fraction[11]. The identification of GerQ filament in this dataset suggest that these filaments are localized outside the bacteria cells or spores. In this study, we observed GerQ filaments in samples without filament extraction, providing further confirmation of the extracellular localization of GerQ filaments. However, to the best of our knowledge, GerQ has not been reported to locate outside the spores or vegetative cells. In the case of *B. subtilis*, GerQ is known to be expressed exclusively in the mother cell compartment of sporulation cells[12]. GerQ-GFP fusion proteins have been observed as dots close to the developing forespore early in sporulation and later assemble around the periphery of the spores during spore development[12]. Another study using fluorescent GerQ fusion and high resolution fluorescence microscopy suggested that GerQ is located in the inner spore coat of *B. subtilis*[18]. Similarly, in *B. cereus* and *B. thuringiensis*, YwdL (a homolog of GerQ) was reported to exclusively localize on the inner surface of the exosporium, an additional outer layer of the spore coat in some *Bacillus* groups[13]. Therefore, it is likely that the extracellular GerQ filaments we observed in this study resulted from leakage of damaged or undeveloped spores. However, we cannot exclude the possibility that these filaments represent a previously unreported localization and potentially an alternative function of GerQ outside the spores and vegetative cells. It is worth noting that these extracellular GerQ filaments were not observed under other culture conditions or in other *Bacillus* bacteria, including germinated *B. amyloiquefaciens* spores, indicating that this leakage or alternative function might be specific to certain situations. Nevertheless, the observation that most residues involved in the nine-subunit assembly of GerQ, layer stacking of the nine-subunit assemblies, and the GerQ-CwlJ interaction are conserved across the *Bacillus* group suggest that the assembly of GerQ and its potential interaction with CwlJ within the spores could be a general mechanism in *Bacillus* bacteria (Supplementary Fig. 4).

To investigate whether GerQ located within the spores forms filaments, we attempted to break purified spores and search for released GerQ filaments. However, we did not observe GerQ filaments in homogenized spores (Fig. 3). This could be due to the fact that although homogenization can partially crack the spores, it might not be sufficient to release the GerQ filaments. We also tried sonication, but obtained similar results. Furthermore, we employed methods that are strong enough for complete spore decoating, such as heat treatment with SDS and low-temperature grinding, but still did not observe GerQ filaments. Under these conditions, it is possible that GerQ filaments may also be destroyed, considering the packing of the filament, especially the layer-to-layer stacking, is not very stable. Therefore, it is challenging to confirm the assembly state of GerQ within the spores at

this current stage. The extracellular location of GerQ filaments, whether caused by accidental leakage or intentional relocation, provides us with an opportunity to study the molecular architecture of the GerQ assembly. Additionally, several studies have reported connections between protein filaments and spores: a fibrous structure has been observed in the inner coat of *B. subtilis* spores[22]; recombinant spore coat proteins have been shown to form filaments[23]; protein filaments have been identified as spore appendages[8]. These findings support the notion that protein filaments are associated with spores and further emphasize the importance of studying GerQ assembly in understanding spore biology.

GerQ is known to form high-molecular-mass cross-links mediated by Tgl, a spore-associated transglutaminase[24]. The first three lysine residues from the N-terminus (Lys2, Lys4, and Lys5) have been identified as responsible for this cross-linking[25]. However, the specific binding partner of GerQ cross-linking remains still unknown, and this cross-linking does not appear to influence GerQ's function, including its role in facilitating spore germination and recruiting CwlJ[24,25]. In our filament structure, we observed that the N-terminal region of GerQ is perturbed outside the filament core and appears flexible without visible densities (Fig. 2f, Supplementary Fig. 3c). It is worth noting that the N-terminal region is less conserved across the *Bacillus* group compared to the filament-forming C-terminal region (Supplementary Fig. 4). Importantly, deletion of 58 residues at the C-terminal, but not 23 residues at the N-terminal, abolishes GerQ function[12,25]. These findings indicate that the filament core, but not the flexible N-terminal region, is responsible for GerQ's function. In the cryo-EM map, we observed connected densities between the side chains of Met91 of one subunit and Arg104 of the adjacent subunit (Supplementary Fig. 3c). However, we consider it less likely that this represents a cross-linking interaction between GerQ subunits because the cross-linking between the side chains of arginine and methionine is not chemically favorable.

In summary, our study identified GerQ filaments in a cryo-EM dataset obtained from *B. amyloiquefaciens* biofilms. Through structure analysis, subsequent experiments, and interaction predictions, we uncovered that GerQ likely recruits CwlJ by forming a bowl-like scaffold, enabling its role in spore germination. We acknowledge that the primary limitation of this study is the absence of experimental evidence regarding GerQ filament formation within the spores and the interaction between GerQ and CwlJ. Consequently, the direct interaction between GerQ and CwlJ, as well as the functional insights into GerQ, are yet to be established experimentally. Nevertheless, this discovery of previously unreported protein filaments in *Bacillus* bacteria contribute valuable insights for understanding sporulation and germination processes. Our findings shed light on the molecular mechanisms underlying these fundamental biological processes.

## Methods

### Cryo-EM dataset re-processed in this study
Cryo-EM dataset was collected in a previously reported study[11]. Briefly, to obtain the cryo-EM data, *B. amyloiquefaciens* were grown on YESCA agar plates (1 g/L yeast extract, 10 g/L casamino acids, 20 g/L agar) and incubated at 25°C for 3 days. Filaments were extracted from the harvested bacteria using homogenization and centrifugation. Cryo-EM grids were prepared using Quantifoil 1.2/1.3 200 mesh electron microscope grids (catalog number N1-C14nCu20-01) and a Vitrobot Mark IV (Thermo Fisher Scientific). Cryo-EM data were collected on a Titan Krios transmission electron microscope (Thermo Fisher Scientific).

### Cryo-EM data processing
Micrographs were processed in cryoSPARC 4.3.0 using the "in-house" Patch Motion Correction and Patch CTF Estimation jobs[26]. Nearly 300 GerQ filament particles were initially picked by hand using the manual picker job and a box size of 360 × 360 pixels. The particles were then

subjected to 2D classification and the good classes were used as inputs for both template picker and filament tracer jobs. In general, we found that since many of the GerQ filaments were closely bundled together the template picker job type performed better than filament tracer. The separation distance between particles was set at 30 Å. The subsequent inspect particle picks job was used to remove only the worst particle types such as cubic ice and carbon but otherwise the thresholds used to select the particles at this stage were not very stringent. Next, particles were extracted using a box size of 764 × 764 pixels binned to 384 × 384 pixels. These large particles were subjected to 2D classification which revealed many classes which had bundles of GerQ filaments. The particles from a 2D class with only a single non-bundled filament were then used for calculation of the power spectrum. The power spectrum was analyzed and possible symmetries were determined as previously described[27–29]. The possible symmetries were quite limited due to the layer lines present (Fig. 1c). First, there was a layer line at -1/(34 Å) with meridional intensity that appeared to be of Bessel order $n = 0$. Second, there was a layer line at -1/(70 Å) that appeared to be of Bessel order greater than or equal to -7 and less than or equal to -13. This pattern suggested that the filament had a point group symmetry (C7, C8, … C13) and that the layer line at -1/(70 Å) contained two Bessel functions of equal order but opposite sign.

To maximize the number of non-bundled particles, particle extraction was performed again using a box size of 512 × 512 binned to 256 × 256. Subsequently, 2D classification of these particles was performed and a single good class of 17,798 particles was selected. Using helical reconstruction in cryoSPARC as performed previously[30], the various possible symmetries were tested until a correct symmetry with clear secondary structure was determined to -3.5 Å. For each symmetry a featureless cylindrical starting volume was used. The correct symmetry had $C_9$ point group symmetry with a rise of 34.2 Å and a twist of −18.4°. Local CTF refinement was then performed and the subsequent particles were input into another helical refinement job with the correct symmetry to reach a slightly higher resolution of 3.3 Å using the "gold standard" 0.143 map:map FSC (Supplementary Fig. 2). We note that we did not provide the representation of the angular distribution of particles used in the final reconstruction, because the reconstruction was conducted using a helical symmetry and a $C_9$ point group symmetry. With $C_9$ symmetry applied, only angles from 0-40° are required for reconstruction. Consequently, we feel that showing such a distribution graphic would be misleading.

To explore the potential binding of CwlJ at the tip of GerQ filaments within our dataset, we manually picked the particles located at the ends of GerQ filaments, yielding a total of 400 particles picked from 1400 micrographs. Since we expect that CwlJ would be bound at only one end of each GerQ filament, this would be a potential 200 CwlJ-GerQ complexes from 1400 micrographs. Further, these tips would lack the 9-fold symmetry that we were able to apply to the GerQ filaments. Due to the limited number of particles and the absence of symmetry, it was impossible to construct a reliable cryo-EM map, eliminating our ability to substantiate the binding of CwlJ to the GerQ filament through cryo-EM structure determination.

### Atomic model building

The final volume was used as input into ModelAngelo 1.0[14] with no given sequence. The program returned a model with multiple subunits which fit the density map well. The sequence from a single subunit from this model was taken and input into a BLAST search[15,16] which output numerous hits for the protein GerQ from various species of *Bacillus* with the top hits being GerQ from *B. subtilis* (66% sequence identity) and GerQ from *B. amyloiquefaciens* (65% sequence identity). The final model was built using COOT v0.9.8.2[31] with the genotyping sequence and was refined with phenix.real_space_refine v.1.20.1-4487[32]. The final model contains two layers of the nine-subunit assembly. The final model was validated using MolProbity[33].

### Genotyping

Genotyping was performed as previously described[11]. A single colony of *Bacillus amyloiquefaciens* was picked from an LB plate and suspended in 200 μL of water. Primers were designed to amplify GerQ encoding gene. PCR amplification was performed using Phusion High-Fidelity DNA Polymerase (New England Biolabs, catalog number M0531S), and the PCR products were sequenced by Shanghai Sangon Biotech Co. The genotyping result is listed in Supplementary Table 1.

### Bacteria culture

To investigate the existence and location of GerQ filament, *B. amyloiquefaciens* were cultured under the same condition as reported previously[11]. A frozen stock of *B. amyloiquefaciens* streak onto a Luria-Bertani (LB) agar plate and incubated overnight at 37 °C. After incubation, a single colony was inoculated into LB broth and cultured overnight at 37 °C. The culture was diluted to an optical density ($OD_{600}$) of 0.01 and shaken at 37 °C for 30 min. Next, 80 μL of the diluted culture was spotted onto YESCA agar plates (1 g/L yeast extract, 10 g/L casamino acids, 20 g/L agar) and incubated at 25 °C for up to 30 days. To investigate whether GerQ fibrils were also present around other *Bacillus* bacteria, *B. subtilis* strain 168 was cultured in YESCA agar plates using the same protocol.

To purify spores for further study, *B. amyloiquefaciens* were cultured in 2 × SG liquid medium as described previously[19]. A single colony of *B. amyloiquefaciens* was inoculated into LB broth and cultured overnight at 37 °C. The incubated culture was mixed with 2 × SG liquid medium (16.0 g/L nutrient broth (Beijing lablead biotech., LTD., catalog number LM1168B), 2.0 g/L KCl, 0.5 g/L $MgSO_4·7H_2O$, 1.0 mL/L 1 M $Ca(NO_3)_2$, 1.0 mL/L 0.1 M $MnCl·4H_2O$, 100 μL/L 10 mM $FeSO_4$, 2.0 mL/L 50% (w/v) glucose) with a ratio of 1:100 (v/v). The mixture was incubated at 37 °C with 200 rpm shaking for 2 days.

### Negative stain TEM imaging of GerQ fibrils

Bacteria cultured under various conditions for different durations were gently scrapped off from the plates, mixed with buffer containing 10 mM Tris·HCl, pH 7.4 and directly applied to glow-discharged 200 mesh carbon coated copper grids (Beijing Zhongjingkeyi Technology Co., Ltd., catalog number BZ11022a) for TEM sample preparation. For bacteria grown in 2 × SG liquid medium, liquid culture was directly applied to the grids. Grids were stained with 2% uranyl acetate and imaged using a Talos L120C G2 transmission electron microscope (Thermo Fisher Scientific). The purified spores, with or without homogenization, were imaged by TEM with the same protocol.

### Spore purification

Spore purification was carried out following a previously established protocol[19]. *B. amyloiquefaciens* cultured in 2× SG liquid medium was harvested by centrifugation at 10,000 × g for 10 min. The supernatant was discarded, and the pellet was washed with sterile water at a volume equal to half of the medium volume, followed by centrifugation at 10,000 × g for 10 min. This washing step was repeated three times. The resulting pellet was then resuspended with sterile water at a volume equal to half of the medium volume and gently shaken at 4 °C overnight. On the following day, the suspension was centrifuged again at 10,000 × g for 10 min, and the pellet was resuspended in sterile water and shaken at 4 °C overnight. This process was repeated for a total of three days. After removing the viscous contaminating layer, the pellet was resuspended in sterile water and centrifuged at 10,000 × g for 10 min. The resulting pellet, consisting predominantly of pure spores, was resuspended in 2 mL of sterile water. The suspension was divided into 1.5 mL Eppendorf tubes in 200-μL aliquots and centrifuged at 10,000 × g for 10 min to remove the supernatant. The purified spores stored at −80 °C for further use.

To assess whether homogenization could disrupt the spores and release GerQ filaments, the purified spores were mixed with a

homogenization buffer (10 mM Tris-HCl, pH 7.4). The mixture was homogenized on ice using a handheld homogenizer for 10 min. Following homogenization, the solution was collected for malachite green staining and negative stain TEM observation. For low-temperature grinding, the purified spore pellet was combined with grinding beads and flash-frozen using liquid nitrogen. The frozen mixture was then subjected to grinding using a Tissuelyser tissue grinder (Jingxin, Shanghai) operating at 50 Hz for 3 min. The ground sample was resuspended in the same buffer used in homogenization (10 mM Tris-HCl, pH 7.4) and used for negative stain EM observation.

## Malachite green staining

A suspension of the bacterial culture was applied to a slide and allowed to air dry. Subsequently, a few drops of 5% malachite green stain were added to the slide. The slide was gently heated over a flame to allow the staining solution to steam for 5 min. Excess staining solution was carefully poured off from the slide. Once the slide had cooled down, it was washed with water until the effluent water became colorless. The slide was then counterstained with a 0.5% safranin solution for 2 min, followed by another water wash. Finally, the slide was dried and examined using an Olympus CX43 microscope under the oil immersion lens at a magnification of 1000×.

## Spore germination

Purified *B. amyloiquefaciens* spores were baked at 70 °C for 3 h to eliminate vegetative cells. The baked spores were resuspended with sterile water to achieve an $OD_{600}$ equal to or greater than 40. This spore suspension was then heat-activated at 70 °C for 30 min. Subsequently, the activated spore solution was mixed with 2× YT medium supplemented with 10 mM L-alanine and optionally 1 mg/ml of 2,3,5-triphenyltetraolium chloride (TTC) and 5 mM glucose at a 1:100 (v/v) ratio. The mixture was incubated at 37 °C or 4 °C for durations of 4 h or 24 h. Spore germination was monitored by assessing the color change of the culture, as germinated spores exhibit red due to the reduction of TTC. The cultures containing germinated spore were directly applied onto EM grids for the observation of GerQ filaments.

## Resin-embedded ultrathin section of purified spores

Purified *B. amyloiquefaciens* spores were initially fixed with 2.5% (v/v) glutaraldehyde at 4 °C overnight. The fixed solution was centrifuged at 5000 × *g* for 5 min, and the supernatant was discarded. The pellet was resuspended in 0.1 M phosphate buffer (PB) and incubated for 15 min, followed by a centrifugation step at 5000 × *g* for 3 min. This washing procedure was repeated four times. The washed pellet was resuspended with 0.2 mL of 1% (v/v) osmic acid and incubated for 1.5 h. After centrifugation at 5000 × *g* for 3 min, the supernatant was discarded, and the pellet was washed for three additional times following the same process described above. The resulting pellet was sequentially dehydrated using 30% (v/v) ethanol, 50% (v/v) ethanol, 70% (v/v) ethanol, 90% (v/v) ethanol, a 1:1 (v/v) mixture of 90% (v/v) ethanol and 90% (v/v) acetone, and finally 90% (v/v) acetone. During each dehydration step, the suspension was incubated for 15 min and centrifuged at 5000 × *g* for 3 min to collect the pellet. Then the pellet was further dehydrated with 100% acetone for 10 min and centrifuged at 5000 × *g* for 3 min. The 100%-acetone dehydration step was repeated three times. After dehydration, the pellet was sequentially incubated with 0.3 mL of a mixture of acetone and resin (Ted Pella, USA, catalog number GP18010). The volume ratios of acetone to resin in each incubation step were 1:1, 1:2, and 1:3, with respective incubation time of 1.5 h, 1 h, and overnight. After each incubation, the solution was centrifuged at 5000 × *g* for 5 min to collect the pellet. Subsequently, the pellet was resuspended with 0.2 mL of resin and incubated for 3 h, followed by centrifugation at 5000 × *g* for 5 min. This step was repeated once, after which 0.1 mL of supernatant was removed. The remaining solution was baked at 60 °C for 2 days.

The resulting resin-embedded sample was processed for ultra-thin sectioning using a Leica EM TP (Leica Microsystems, Germany). The ultra-thin slices were applied to EM grids, which were then stained with a uranium-free dye (Ted Pella, USA, catalog number 19485) for 10 min and blotted dry using filter paper. Subsequently, the grids were stained with lead citrate for 6 min, followed by blotting with filter paper. The grids were air-dried for 2 min and imaged using a Talos L120C G2 transmission electron microscope (Thermo Fisher Scientific).

## Molecular docking and AlphaFold 3 prediction

Macromolecular docking was performed using the LightDock server (https://server.lightdock.org/) and HDock server (http://hdock.phys.hust.edu.cn/).The cryo-EM structure of GerQ, which consists of two layers of the nine-subunit assembly (a total of 18 GerQ subunits), was used as the receptor in the docking simulation. As for the ligand, the *B. amyloiquefaciens* CwlJ structure predicted using the AlphaFold 3 server (https://golgi.sandbox.google.com/) was employed. The predicted model of GerQ-CwlJ complex was generated using the AlphaFold 3 server, with the sequences of *B. amyloiquefaciens* CwlJ and the genotyped *B. amyloiquefaciens* GerQ provided. Nine copies of GerQ and one copy of CwlJ was requested for AlphaFold 3 prediction. The sequences used for the AlphaFold 3 prediction were listed in the source data file.

## Reporting summary

Further information on research design is available in the Nature Portfolio Reporting Summary linked to this article.

## Data availability

Cryo-EM map and atomic model of *Bacillus amyloiquefaciens* GerQ have been deposited into the Worldwide Protein Data Bank (wwPDB) and the Electron Microscopy Data Band (EMDB) with accession codes PDB 8ZRA and EMD-60394. Any other relevant data are available from the corresponding authors upon request. Source data are provided with this paper.

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

## Acknowledgements

We thank H. Yang from Lanzhou University for sharing the *Bacillus amyloiquefaciens* bacteria. This work was supported by STI2030-Major Projects 2022ZD0212500 to Q.C. and the National Natural Science Foundation (NSF) of China (grant nos. 32271276) to Q.C., and by NIH GM122510 to E.H.E. The authors thank the Instrument Analysis Center (IAC), Shanghai Jiao Tong University, for cryo-EM data collection. The authors acknowledge the National Facility for Translational Medicine (Shanghai) for support.

## Author contributions

M.K. and E.H.E. processed the cryo-EM data. M.K., Y.C., J.H., and Q.C., built the atomic model. Y.C. performed genotyping, bacteria culture, spore purification, malachite green staining, and negative stain TEM observation. Y.C. and Q.C. performed molecular docking and AlphaFold 3 prediction. All authors analyzed the results and wrote the manuscript. E.H.E. and Q.C. supervised the project.

## Competing interests

The authors declare no competing interests.
