## [Peer Review File · Nature Communications]

Molecular architecture of the assembly of *Bacillus* spore coat protein GerQ revealed by cryo-EMREVIEWER COMMENTS

Reviewer #1 (Remarks to the Author):

The manuscript by Cheng et al. describes filaments extracted from *Bacillus amyloiquefaciens*. This bacterium forms biofilms, which consist of many different proteins. The extracted filaments studied here, consist of GerQ, a coat protein associated with germination. It was so far not known that GerQ self-assembles into filaments.

The authors could also show that GerQ filaments are found outside spores, but could not find evidence, that they are also present inside the spores. Since the appearance of GerQ filaments coincides with the emergence of spores, these filaments might be associated with the sporulation process. It was not known that GerQ filaments can be found outside the spores.

GerQ is known to interact with CwJ, which is essential for spore germination, but it is unclear whether CwJ directly binds to GerQ.

The authors performed molecular docking of their GerQ filament structure with an AlphaFold3 model of CwJ and found a potential binding site inside the bowl-like structure GerQ, further supported by the fact that the AlphaFold3 complex prediction of GerQ with CwJ found the same arrangement.

The work seems to be well done. The text is well written and the figures are very clear. The results are very interesting and provide new insight into the structure and function of GerQ and its interaction with CwJ, which is relevant for better understanding the germination process. I do not have any other comments.

typos:

line 37: "it is composition" -> "its composition"

line 39: "hash" -> "harsh"

line 40: "these" -> "there"

line 109: "continues" -> "continuous"

line 417: "LighDock" -> "LightDock"

Reviewer #2 (Remarks to the Author):

1. What are the noteworthy results? The CryoEM structure of GerQ filaments presented is certainly novel, and the docking of the CwJ protein to the top of the GerQ filament indicates how CwJ may assemble in spores' outer layers, and results are consistent with the interdependence of GerQ and

CwJ in spore coats. The CryoEM work overall appears well done. However, docking CwJ to the GerQ nine-subunit ring is not sufficient. Given the authors' ability to resolve the 8-nm GerQ single filaments, they should attempt to identify CwJ at the ends of the GerQ filaments in the electron micrographs on material from biofilms.

2. Will the work be of significance to the field and related fields? The work is certainly original and adds another brick in the wall that is the *Bacillus* spore coat, and the work is well done. The one "gap" in the knowledge is whether GerQ actually forms filaments in the spore coat, as there is no evidence that this is the case, although the authors have tried to look for them. However, they could have been more "creative" in doing this, such as looking in germinated spores, and perhaps in spores making defective coats. Consequently, there is no definitive knowledge of the function of the GerQ filament in spores, and thus its physiological relevance is uncertain, other than how GerQ "might" recruit CwJ in spores.

3. Does the work support the conclusions and claims, or is additional evidence needed? As noted above in #s1 and 2, while the structural results from the filaments in the *B. amyloliquefaciens* biofilms are solid, there is no evidence that this filamentous structure is actually present in spores themselves even though this seems likely. However, the docking of CwJ to the "bowl-like" structure formed by the GerQ filaments certainly indicates how GerQ-CwJ association may take place, although as noted in #1, it should perhaps have been possible to see if CwJ was at the ends of the GerQ filaments from the biofilms. (Note here that in multiple places in the ms, "bowl-like" has been written incorrectly as "bow-like").

4. Are there any flaws in the data analysis, interpretation and conclusions? Do these prohibit publication or require revision? As noted above, currently there is no evidence that the filamentous GerQ structure with CwJ in the "bowl" is the way these proteins are assembled in spores' coat. I do not believe this prohibits publication, but the authors need to make this issue clear, and as noted in 2 above, could have been more creative in looking for GerQ filaments in spores!

5. Is the methodology sound? Does the work meet the expected standards in your field? Yes and Yes.

6. Is there enough detail provided in the methods for the work to be reproduced? Yes.

7. There are multiple typos in the ms, some of which are the following.

Line 109: "continues" -> "continuous."

Line 117: "bow" -> "bowl." This error is present in numerous lines as noted above.

Line 122: "Ty98" -> "Tyr98."

Line 323: "point a group" -> "a point group."

Line 396: "200 μ L" -> "200- μ L."

Line 397: "10,000 g" -> "10,000x g."

Line 417: "LighDock" -> "LightDock." This error is also present in many lines.

Reviewer #3 (Remarks to the Author):

We thank the reviewers for the valuable suggestions of our manuscript, which has played a crucial role in enhancing the quality of our research. In response to the reviewers' suggestions, we have made the following modifications:

1. We have added a description of the manual picking of the filament ends in the Method section, as recommended by reviewer #2.
2. To address the concerns raised by reviewer #2 in comment #2, we have added three additional experiments aimed at identifying GerQ filament under varied conditions.
3. Other minor revisions according to reviewers' suggestions.

All changes are highlighted in revised manuscript, and our point-by-point responses to the reviewers follow below:

Reviewer #1 (Remarks to the Author):

The manuscript by Cheng et al. describes filaments extracted from *Bacillus amyloiquefaciens*. This bacterium forms biofilms, which consist of many different proteins. The extracted filaments studied here, consist of GerQ, a coat protein associated with germination. It was so far not known that GerQ self-assembles into filaments.

The authors could also show that GerQ filaments are found outside spores, but could not find evidence, that they are also present inside the spores. Since the appearance of GerQ filaments coincides with the emergence of spores, these filaments might be associated with the sporulation process. It was not know that GerQ filaments can be found outside the spores.

GerQ is known to interact with CwlJ, which is essential for spore germination, but it is unclear whether CwlJ directly binds to GerQ.

The authors performed molecular docking of their GerQ filament structure with an AlphaFold3 model of CwlJ and found a potential binding site inside the bowl-like structure GerQ, further supported by the fact that the AlphaFold3 complex prediction of GerQ with CwlJ found the same arrangement.

The work seems to be well done. The text is well written and the figures are very clear. The results are very interesting and provide new insight into the structure and function of GerQ and its interaction with CwlJ, which is relevant for better understanding the germination process.

I do not have any other comments.

Response: We thank the reviewer for the positive and constructive comments regarding our manuscript.

typos:

line 37: "it is composition" -> "its composition"

line 39: "hash" -> "harsh"

line 40: "these" -> "there"

line 109: "continues" -> "continuous"

line 417: "LighDock" -> "LightDock"

Response: We have corrected these typos in the revised manuscript, and we thank the reviewer for pointing them out.

Reviewer #2 (Remarks to the Author):

1. What are the noteworthy results? The CryoEM structure of GerQ filaments presented is certainly novel, and the docking of the CwlJ protein to the top of the GerQ filament indicates how CwlJ may assemble in spores' outer layers, and results are consistent with the interdependence of GerQ and CwlJ in spore coats. The CryoEM work overall appears well done. However, docking CwlJ to the GerQ nine-subunit ring is not sufficient. Given the authors' ability to resolve the 8-nm GerQ single filaments, they should attempt to identify CwlJ at the ends of the GerQ filaments in the electron micrographs on material from biofilms.

Response: We have followed the reviewer's advice and attempted to find evidence of CwlJ-GerQ interaction by manually picking particles solely on the ends of GerQ filaments in our cryo-electron micrographs. Regrettably, we were only able to isolate 400 ends of the GerQ fibrils from a total of 1,400 of our micrographs, significantly below the requisite particle count for determining the structure of the filament tips. Meanwhile, our hypothetical model has CwlJ exclusively binding to one end of each polar GerQ filament, so we would have obtained 200 CwlJ ends from 1,400 micrographs, and 200 naked ends. In this study, a 3.5-Å cryo-EM structure was derived from the relatively small number of 17,798 GerQ particles, possible only because of the nine-fold symmetry of the GerQ filament. In our proposed GerQ-CwlJ complex model, CwlJ binds at the center of the nine-subunit GerQ assembly with a 1:9 stoichiometry, thereby eliminating the possible application of the nine-fold symmetry in the structure determination of the GerQ-CwlJ complex. Consequently, a substantially larger particle count is essential to attain near-atomic resolution. To illustrate, assuming optimistically that a minimum of 20,000 particles are required to achieve a 4-Å resolution asymmetric structure of the GerQ filament tip, the data collection process would necessitate 140,000 micrographs under our current experimental conditions. This extensive volume of images poses challenges in terms of cost and storage capacity.

We believe that the primary objective of this manuscript is to present a groundbreaking discovery regarding the role of GerQ as a protein in *Bacillus amyloiquefaciens*. Our study demonstrates the polymerization of this protein into filaments, marking a significant advancement in our understanding of GerQ. While we acknowledge the speculative nature of the docking studies involving CwlJ, given the limited existing literature on GerQ, we consider the level of speculation in our paper regarding this interaction to be warranted. Irrespective of the outcomes of future research on GerQ filaments and their interactions with other proteins, we are confident that the findings presented in this study will play a crucial role in unraveling the functions of protein complexes in *Bacillus* sporulation moving forward. We believe that future investigations delving into this interaction may prove more feasible and fruitful and looking at a CwlJ-GerQ complex is beyond the scope of the present study.

We have included a description of the manual picking of the filament ends in the Method and Discussion section of the revised manuscript.

2. Will the work be of significance to the field and related fields? The work is certainly original and adds another brick in the wall that is the *Bacillus* spore coat, and the work is well done. The one “gap” in the knowledge is whether GerQ actually forms filaments in the spore coat, as there is no evidence that this is the case, although the authors have tried to look for them. However, they could have been more “creative” in doing this, such as looking in germinated spores, and perhaps in spores making defective coats. Consequently, there is no definitive knowledge of the function of the GerQ filament in spores, and thus its physiological relevance is uncertain, other than how GerQ “might” recruit CwlJ in spores.

Response: We appreciate the reviewer’s favorable assessment of the significance of our research. In addressing the “gap” highlighted by the reviewer, we have followed the reviewer’s advice and conducted experiments to detect GerQ filaments under three additional conditions: i) around germinated spores; ii) post-application of low-temperature grinding to disrupt purified spores; iii) within ultrathin sections of spore coats. Regrettably, no GerQ filaments were observed under these experimental conditions. Nevertheless, we maintain that these results do not necessarily negate the presence of GerQ filaments within the spore coat. It is plausible that during germination, GerQ filaments are located inside the spores rather than externally; the force generated by low-temperature grinding might be too intense to disassemble GerQ filaments; and the dense nature of spore coats sections could impede the identification of GerQ filaments. Intriguingly, ring-like particles resembling the bowl-like structure of GerQ were observed in ground spores, offering partial albeit indefinite support of the assembly of GerQ within spores.

These additional experiments, in conjunction with those outlined in the original manuscript, underscore the formidable challenge of definitively confirming the assembly state of GerQ within the spores, a task that may potentially surpass the scope of this study. We trust that the reviewer concurs with our perspective that our current work, by presenting the high-resolution structure of GerQ filament cultivated under nature conditions, puts forth a plausible hypothesis delineating GerQ assembly and functional mechanisms within bacterial spores. We believe that this study will stimulate further exploration in the field to experimentally elucidate the assembly state and functional role of GerQ.

The experiments mentioned above have been integrated into the Results, Discussion, and Method sections of the revised manuscript, along with their representation in Fig. S5.

3. Does the work support the conclusions and claims, or is additional evidence needed? As noted above in #s1 and 2, while the structural results from the filaments in the *B. amyloliquefaciens* biofilms are solid, there is no evidence that this filamentous structure is actually present in spores themselves even though this seems likely. However, the docking of CwlJ to the “bowl-like” structure formed by the GerQ filaments certainly indicates how GerQ-CwlJ association may take place, although as noted in #1, it should perhaps have been possible to see if CwlJ was at the ends of the GerQ filaments from the biofilms. (Note here that in multiple places in the ms, “bowl-like” has been written incorrectly as “bow-like”).

Response: Please see our response to the comment #1 of reviewer #2.

4. Are there any flaws in the data analysis, interpretation and conclusions? Do these prohibit publication or require revision? As noted above, currently there is no evidence that the filamentous GerQ structure with CwlJ in the “bowl” is the way these proteins are assembled in spores’ coat. I do not believe this prohibits publication, but the authors need to make this issue clear, and as noted in 2 above, could have been more creative in looking for GerQ filaments in spores!

Response: we agree with the reviewer that currently there is no experimental evidence of the assembly state and functional mechanism of GerQ. This acknowledgment has been underscored within our manuscript, particularly in the Discussion section, as exemplified in the second paragraph: “However, it is important to note that the interaction between GerQ and CwlJ is currently hypothetical and lacks experimental evidence.....Further experimental studies are required to validate the interaction of GerQ and CwlJ, as well as to confirm the assembly states of GerQ within the spores.”

5. Is the methodology sound? Does the work meet the expected standards in your field? Yes and Yes.

Response: We thank the reviewer for the positive feedback regarding our manuscript.

6. Is there enough detail provided in the methods for the work to be reproduced? Yes.

Response: We thank the reviewer for the positive feedback regarding our manuscript.

7. There are multiple typos in the ms, some of which are the following.

Line 109: "continues" -> "continuous."

Line 117: "bow" -> "bowl." This error is present in numerous lines as noted above.

Line 122: "Ty98" -> "Tyr98."

Line 323: "point a group" -> "a point group."

Line 396: "200 μ L" -> "200- μ L."

Line 397: "10,000 g" -> "10,000x g."

Line 417: "LighDock" -> "LightDock." This error is also present in many lines.

Response: We are sorry for these typos, and we have corrected them in the revised manuscript.

Reviewer #3 (Remarks to the Author):

Response: We sincerely thank the reviewer for the feedback which has helped to improve the quality of our manuscript.

REVIEWER COMMENTS

Reviewer #2 (Remarks to the Author):

The authors have corrected typos in the original submission, and now have pointed out the unknowns still remaining about their findings, specifically the lack of evidence that: i) there is indeed interaction between CwJ and GerQ consistent with their docking data; and ii) the filaments found in biofilms are actually present in spores. They also describe their efforts to resolve the two unknowns noted above, unfortunately without success. Consequently, there are still concerns about whether the two seminal findings in the manuscript reflect the properties/arrangement of these two proteins in spores themselves, although they certainly may.

In addition, it seems likely that new experimental approaches will be needed to resolve the major remaining unknowns. Possible experimental approaches include:

1. Lack of Experimental Evidence for Assembly State and Functional Mechanism of GerQ

It seems that both the N- and C-termini of GerQ are exposed on the surface of the filaments. The authors could consider fusing GerQ with a fluorescent protein such as GFP or mEOS2 through gene targeting and then using confocal or super-resolution microscopy (e.g., FPALM or STORM) to observe if this occurs in vivo.

2: No Evidence of Interaction Between GerQ and CwJ

To address this, the authors could label GerQ and CwJ with different fluorescent proteins (e.g., mEGFP for GerQ and mCherry for CwJ) and use Total Internal Reflection Fluorescence (TIRF) microscopy to examine whether GerQ filaments interact with the end-binding protein CwJ. Perhaps an appropriate FRET experiment might be helpful here also.

Reviewer #3 (Remarks to the Author):

We thank the reviewers again for the valuable suggestions. Regrettably, we are unable to conduct the fluorescence-based experiments suggested by Reviewer #2 at the current stage. Instead, we have revised the manuscript to include additional discussions that further emphasize the primary limitation of this study, which is the absence of experimental evidence of GerQ filament formation within the spores and the direct interactions between GerQ and CwlJ. The revisions are highlighted in yellow, while similar statements already included in original manuscript are highlighted in blue. Please find our point-by-point responses to the reviewers below:

Reviewer #2 (Remarks to the Author):

The authors have corrected typos in the original submission, and now have pointed out the unknowns still remaining about their findings, specifically the lack of evidence that: i) there is indeed interaction between CwlJ and GerQ consistent with their docking data; and ii) the filaments found in biofilms are actually present in spores. They also describe their efforts to resolve the two unknowns noted above, unfortunately without success. Consequently, there are still concerns about whether the two seminal findings in the manuscript reflect the properties/arrangement of these two proteins in spores themselves, although they certainly may.

In addition, it seems likely that new experimental approaches will be needed to resolve the major remaining unknowns. Possible experimental approaches include:

1. Lack of Experimental Evidence for Assembly State and Functional Mechanism of GerQ

It seems that both the N- and C-termini of GerQ are exposed on the surface of the filaments. The authors could consider fusing GerQ with a fluorescent protein such as GFP or mEOS2 through gene targeting and then using confocal or super-resolution microscopy (e.g., FPALM or STORM) to observe if this occurs in vivo.

2: No Evidence of Interaction Between GerQ and CwlJ

To address this, the authors could label GerQ and CwlJ with different fluorescent proteins (e.g., mEGFP for GerQ and mCherry for CwlJ) and use Total Internal Reflection Fluorescence (TIRF) microscopy to examine whether GerQ filaments interact with the end-binding protein CwlJ. Perhaps an appropriate FRET experiment might be helpful here also.

Response: We agree with the reviewer that the primary limitation of this study lies in the absence of evidence regarding the interaction between CwlJ and GerQ, as well as the formation of GerQ filaments within the spores. We appreciate the reviewer's suggestion of conducting fluorescence-based experiments to address these two major remaining unknowns. Regrettably, these experiments are outside the scope of expertise of our research group and may not be feasible within a reasonable timeframe. Moreover, the intricate environment of GerQ within the spores presents additional challenges for such experiments, given GerQ's localization in the inner coat of the spores—an encapsulated environment with dimensions of approximately a couple of micrometers in diameter and tens of nanometers in thickness, densely packed with numerous proteins. Therefore, we propose that fluorescent-based studies of GerQ within the spores be pursued as a separate follow-up investigation to delve deeper into the function of GerQ in sporulation and germination. We have included

discussions in the revised manuscript to further underscore the limitations of the current study.